

# A limitation of the Cognitive Reflection Test: familiarity

Stefan Stieger[1,2] and Ulf-Dietrich Reips[1]

[1] Department of Psychology, University of Konstanz, Konstanz, Germany
[2] School of Psychology, University of Vienna, Vienna, Austria

## ABSTRACT

The Cognitive Reflection Test (CRT; *Frederick, 2005*) is a frequently used measure of cognitive vs. intuitive reflection. It is also a frequently found entertaining 'test' on the Internet. In a large age-stratified community-based sample ($N = 2,272$), we analyzed the impact of having already performed the CRT or any similar task in the past. Indeed, we found that 44% of participants had experiences with these tasks, which was reflected in higher CRT scores (Cohen's $d = 0.41$). Furthermore, experienced participants were different from naïve participants in regard to their socio-demographics (younger, higher educated, fewer siblings, more likely single or in a relationship than married, having no children). The best predictors of a high CRT score were the highest educational qualification (4.62% explained variance) followed by the experience with the task (3.06%). Therefore, we suggest using more recent multi-item CRTs with newer items and a more elaborated test construction.

## INTRODUCTION

The Cognitive Reflection Test (CRT) was introduced by *Frederick (2005)* and is supposed to be a measure of cognitive reflection in contrast to intuition. It consists of three mathematical/numerical text-based problems, which elicit first an intuitive (wrong) answer and can only be solved when consciously thinking of the (not obviously) true answer. The theory behind this task assumes that there are two distinct cognitive processes: a fast intuitive one and a slow and rather reflective one (*Epstein, 1994*). Some researchers called them System 1 (i.e., spontaneous, instantly, effortlessly) and System 2 processes (i.e., effortful, motivated, reflected; *Stanovich & West, 2000*). To solve the CRT items, one has to ignore the first intention of the System 1 processes and switch to System 2 processes to think intentionally about the correct answer. The CRT has been frequently used and within many research topics (e.g., ideology: *Deppe et al., 2015*; superstitious and paranormal beliefs: *Pennycook et al., 2012*).

The CRT has also been frequently studied from a methodological point of view. One frequently researched topic is whether the CRT is rather a measure of numerical ability than of cognitive reflection. Using a mathematical modeling approach, *Campitelli & Gerrans (2014)* found that the CRT is not only a measure of numerical ability, but also of rational thinking and open-minded thinking for males and of mathematical ability and

Corresponding author
Stefan Stieger,
stefan.stieger@uni-konstanz.de

rational thinking for women. *Sinayev & Peters (2015)* analyzed this issue in a large-*N* multi-study design and found that numerical ability seems to be the key mechanism behind the CRT score (for a similar reasoning, see *Welsh, Burns & Delfabbro, 2013*). Furthermore, although the CRT could prove its usefulness in predicting normative decision making in (primarily) laboratory tasks (*Toplak, West & Stanovich, 2014*), it failed to predict real-life decision outcomes when controlling for personality and decision-making styles (*Juanchich et al., 2016*).

Some researchers suggested an alternative score of intuition (in contrast to reflection; e.g., *Brosnan et al., 2014*), but recent research found that the reflective score still predicts behavior (e.g., intuitive-analytic cognitive styles) better than the intuitive score (*Pennycook et al., 2016*). Because the CRT has only three items, it often lacks high reliability values (range between 0.60 and 0.74; *Weller et al., 2013*; *Liberali et al., 2012*; for a short review, see *Campitelli & Gerrans, 2014*). Therefore, multi-item revisions of the CRT with higher reliability values have been introduced (e.g., seven items: *Toplak, West & Stanovich, 2014*; five items: *Gòmez-Chacòn et al., 2012*), some with psychometric properties scrutinized by using Item Response Theory (six items: *Primi et al., 2015*), which come with the additional advantage of being less susceptible to floor and ceiling effects. To sum up, although some methodological topics are still under debate, the CRT keeps developing and is widely used.

The CRT's frequent use and constant further development might explain its popularity, the Web of Science lists 386 citations and Google Scholar lists 1,286 citations. All the more, several well-known books cite this task (e.g., *Kahneman, 2011*) and the CRT is a frequently found 'test' on the Internet where people can try to solve the items in a game-like competitive manner with the aim of being better than their friends and/or peers. For example, the search term 'Cognitive Reflection Test' elicits 9,130 search results and the first few words of the first item of the test ('a bat and a ball cost . . .') creates 8,570 hits on the Internet (German version 'ein Schläger und ein Ball . . .' 2,970 hits). Recent reports about people working on Amazon's Mechanical Turk platform (i.e., a microjob web platform also called MTurk; *Buhrmester, Kwang & Gosling, 2011*) state that the CRT items are the ones which are probably most often asked in MTurk projects (http://www.pbs.org/newshour/updates/inside-amazons-hidden-science-factory/).

We found one study that investigated the influence of popularity on CRT results. *Brañas-Garza, Kujal & Lenkei (2015)* performed a meta-analysis about the CRT and also included a cross-temporal meta-analysis, where they correlated the mean CRT score (per year), with the respective year of publication. Overall, they found some support for their assumption that, with time, CRT scores are increasing and this effect seems to be driven by online studies (where participants have the possibility to look up the correct answer) although effect sizes were low. However, in this cross-temporal meta-analysis the moderator variable of having already performed the CRT was not explicitly assessed, rather it was assumed that in later samples participants might have already performed such tasks and are more experienced (*Brañas-Garza, Kujal & Lenkei, 2015*).

The CRT obviously only works well if participants are not aware of the rationale behind the task, i.e., that the CRT items elicit a spontaneous, intuitive answer and that this

answer is not the correct one. Therefore, we asked whether the obvious prominence of the task could have an effect on the CRT's outcome (for a similar reasoning, see *Toplak, West & Stanovich, 2014*). If participants are aware of the rationale, they can consciously overcome their first intuition and think about the items more thoroughly knowing that the correct answer is probably not the first answer that came to mind. If this assumption is correct, then participants who have already performed the CRT (or any other similar task) in the past, should have higher CRT scores than participants doing the CRT for the very first time.

To sum up, we investigated whether having already performed the CRT or any similar task (e.g., multi-item versions of the CRT: *Toplak, West & Stanovich, 2014*) in the past has an effect on the CRT's outcome (mean difference, floor and ceiling effects) and whether differences in the sample composition regarding socio-demographics (e.g., sex, age, highest educational qualification, current relationship status) influence this effect.

## METHOD

### Participants

Participants ($N = 2{,}272$) were recruited by word-of-mouth through friends, relatives, and friends-of-friends of several research assistants following a convenience sampling approach. Participants filled in the questionnaire wherever they were approached (e.g., at home, at the university). The final sample was age-stratified and constitutes German-speaking volunteers (predominantly Austria and Germany) from all walks of life, $M_{age} = 39.8$ years, $SD = 17.7$; 56.6% women. We used six different age-strata (18–25, 26–30, 31–40, 41–50, 51–60, 61+) with the aim of an equal number of participants in each strata in the final sample. In terms of highest educational qualification, 4% had not completed primary education, 15% had completed primary education, 23% had an apprenticeship diploma, 34% had completed secondary education, and 24% had a university degree. Participants' current relationship status was: 24% single, 29% in a relationship, 38% married, 4% divorced, 4% widowed, and 1% stated a different relationship status. Almost half of participants had children (46%), 26% were currently smoker, and had on average 1.8 siblings (Median = 1; range 0–11; 13% were the only child in their family).

### Materials

#### Cognitive Reflection Test (CRT)

The CRT (*Frederick, 2005*) is a measure of cognitive in contrast to intuitive reflection. It contains three problems, which first elicit a spontaneous, but mathematically wrong answer. Only individuals who overcome this first intention and deliberately think about the correct answer can solve the problems (e.g., first problem "A bat and a ball cost $1.10 in total. The bat costs $1.00 more than the ball. How much does the ball cost? _____ cents;" intuitive answer = 10 cents; correct answer = 5 cents). The CRT has no time limit and the total score is calculated as the number of correct answers (range 0–3; for descriptive results, see Table 1).[1] Directly after the three CRT

[1] We have also tried the intuitive score as an alternative (for a recent comparison, see *Pennycook et al., 2016*), but found very little differences in the results. This is probably due to the rather low rate of participants who gave a wrong non-intuitive answer (see Table 1).

**Table 1 Success rate of the CRT items, results from the present study.**

| | Correct answer | Wrong (intuitive) answer | Empirical results (%) | | | Missing | Sum |
|---|---|---|---|---|---|---|---|
| | | | Correct | Wrong (intuitive answer) | Wrong (but not intuitive answer) | | |
| CRT Item 1 | 5 | 10 | 28.3 | 68.1 | 2.3 | 1.3 | 100.0 |
| CRT Item 2 | 5 | 100 | 55.0 | 33.1 | 10.0 | 1.9 | 100.0 |
| CRT Item 3 | 47 | 24 | 53.0 | 34.9 | 8.0 | 4.1 | 100.0 |

Notes:
CRT, Cognitive Reflection Test.
Item 1: "A bat and a ball cost €1.10 in total. The bat costs €1.00 more than the ball. How much does the ball cost? _____ cents."
Item 2: "If it takes five machines 5 min to make five widgets, how long would it take 100 machines to make 100 widgets? _____ min."
Item 3: "In a lake, there is a patch of lily pads. Every day, the patch doubles in size. If it takes 48 days for the patch to cover the entire lake, how long would it take for the patch to cover half of the lake? _____ days."

items, we asked participants whether they had done these tasks or any similar ones before (yes/no).

### Demographics

Participants were asked about their sex, age, highest educational qualification, and current relationship status. Because we were interested whether there are general differences between those participants having already performed the CRT or any similar task in the past and those without experience, we also asked further specific demographics without having any specific research question in mind only to give a more complete picture of potential differences in demographics, i.e., whether or not they have children (yes/no), smoke (yes/no), and how many siblings they have, if any.

### Procedure

Participants gave their informed consent, completed the CRT along with several other measures that were not part of this study, and finally provided demographic details. For the purpose of anonymity, each questionnaire was put into an envelope and thrown into a box. All participants took part on a voluntary basis and were not financially remunerated for participation.

### Dominance analysis

When calculating linear regressions, multicolinearity (i.e., intercorrelation between predictors) is a problem when it turns out to be substantial. To quantify multicolinearity, statistical packages calculate the so-called Variance Inflation Factors (VIFs). In the present study, multicolinearity was foreseeable, because demographic variables usually correlate (e.g., participant age is correlated with the highest educational qualification and having children or not).

To account for multicolinearity, we decided to additionally conduct a dominance analysis (*Azen & Budescu, 2003*; *Budescu, 1993*). Dominance analyses have the advantage of assessing the importance of each predictor relative to the other predictors in the regression model. This is realized by looking at the contribution of each single

predictor in the linear model not only in conjunction with other predictors, but also in isolation. Practically, all possible combinations of predictors are used to calculate partial, direct, and total effect parts by decomposing the total $R^2$ (explained variance). The partial effects come from all possible combinations of predictors on the outcome measure by excluding either one or more predictors from the model. The direct effect is the independent contribution without the other predictors in the model (i.e., zero-order correlation with the outcome measure). The total effect represents the classical multiple linear regression when all predictors are included in the model at once. The outcomes of the dominance analysis are $R^2$ values for each predictor, which are adjusted for shared variances with other predictors (i.e., representing the real explained variance). In the present study, dominance analyses were calculated using the R package 'yhat' (*Nimon & Oswald, 2013*; for a recent application, see *Stieger et al., 2014*).

### Ethics

The present study was conducted in accordance with the principles of the Declaration of Helsinki and with institutional guidelines of the School of Psychology, University of Vienna. Furthermore, the present study followed the Guidelines for ethical conduct of behavioral projects involving human participants proposed by the American Psychological Association. According to the institutional guidelines of the University of Vienna, Austria (http://satzung.univie.ac.at/ethikkommission-der-universitaet-wien/), approval by an ethics committee was not necessary because the study did not affect the physical or psychological integrity, the right for privacy, or other personal rights or interests (see §2(1)). All participants gave verbal informed consent after having received a written description of the study. Data collection was anonymous and no harmful procedures were used.

## RESULTS

### Descriptives

Out of the overall sample of 2,272 participants, 135 (5.9%) did not answer at least one of the three CRT questions (percentage of missing answers, see Table 1) or did not state whether or not they had experience with the test ($n = 23$). Data from these participants were excluded from further analyses, leading to a final sample size of $N = 2,137$. In 45 cases (2.1%), participants misstated an obviously correct answer with the first item of the CRT by ignoring the fact that the result should be stated in Cent not in Euros (e.g., participants stated 0.05 instead of five). These inconsistencies were solved and these cases included. We found no inconsistencies for the second and third item of the CRT. Descriptive statistics of the CRT items can be found in Table 1.

The question whether or not participants had already performed the CRT or any similar task in the past were answered with "Yes" by almost half of participants (44.9%, $n = 959$). We will differentiate between naïve and experienced participants in the following sections. The reliability estimate of the CRT for the experienced participants

was slightly higher than the one for the naïve participants (Cronbach α = 0.657 vs. 0.595, respectively).

## Mean differences: are there differences between naïve and experienced participants?

As can be seen from Table 2, there were several differences between the participant groups, not only regarding the CRT but also regarding several demographic variables. First of all, as hypothesized experienced participants had significantly higher CRT scores than naïve participants (low-to-medium effect size; *Cohen, 1988*). This experience effect affected all CRT items more or less to the same extent (Odds Ratios *OR*s between 1.66 and 2.05).

Furthermore, experienced and naïve participants differed in their demographics. Experienced participants were younger, had a higher education, fewer siblings, no children, and were more likely single or in a relationship than married (see standardized residuals in Table 2).

The difference between experienced and naïve participants regarding the CRT score becomes even more pronounced for different educational levels (see Table 3). Compared to the university samples presented by *Frederick (2005)*, participants in the present study having experience with the CRT or any similar tasks and having an apprenticeship diploma as highest educational level, performed slightly better than Harvard students (1.50 vs. 1.43). The same applies to participants with secondary education, they were similar to Princeton students (1.65 vs. 1.63), and participants with a university degree performed almost as well as Massachusetts Institute of Technology (MIT) students (1.99 vs. 2.18). Furthermore, floor and ceiling effects became relevant. About one third of experienced participants with secondary education or a university degree reached the highest score possible (= 3), whereas again one third of experienced participants with lower education did not solve any CRT item.

In a next step, we calculated a linear regression with the CRT score as the dependent measure to better quantify the influence of the CRT knowledge and participants' demographics onto the CRT score. Because some of the predictors are intercorrelated (e.g., age with current relationship status, highest educational qualification, and having children), multicolinearity can be assumed. This is a problem for linear regressions, because the influence of predictors onto the dependent measure are not pure, they may depend on other predictors. Therefore, we also calculated a dominance analysis. Results can be found in Table 4. VIFs, which indicate multicolinearity, were between 1.034 and 17.692 (following current practices, VIFs higher than 10 are regarded as problematic; *O'Brien, 2007*).

The impact of multicolinearity on the beta weights can also be tested by calculating Spearman rank-order correlations between dominance weights (which should be true values adjusted for intercorrelations) and the absolute values of beta weights. In case of no multicolinearity, order ranks of dominance weights should resemble the order ranks of beta weights (i.e., perfect rank-order correlation of one). If the rank-order correlation deviates from one, the more likely multicolinearity is present. In fact, the rank-order

**Table 2 Differences between experienced and naïve participants regarding the variables under investigation.**

|  | Experienced M (SD) | Naïve M (SD) | t-test | Cohen's d |
|---|---|---|---|---|
| CRT sum score | 1.65 (1.11) | 1.21 (1.06) | 9.38*** | 0.41 |
| Age | 35.0 (16.00) | 43.4 (17.93) | 11.36*** | 0.50 |
| Education | 3.8 (1.12) | 3.5 (1.10) | 7.11*** | 0.31 |
| Number of siblings | 1.7 (1.46) | 1.9 (1.61) | 3.44** | 0.15 |
|  | Standardized residuals |  | $\chi^2$ | OR (CI) |
| CRT Item 1 | −3.0/4.7 | 2.7/−4.2 | 56.25*** | 2.05 (1.70, 2.48) |
| CRT Item 2 | −3.5/2.8 | 2.9/−2.5 | 32.86*** | 1.66 (1.40, 1.98) |
| CRT Item 3 | −4.3/3.9 | 3.9/−3.5 | 61.56*** | 2.01 (1.69, 2.39) |
| Sex (m/f) | 0.8/−0.7 | −0.7/0.6 | 1.97 | 0.88 (0.74, 1.05) |
| Relationship status[1] | 2.7/3.8/−4.8/−0.4/ −2.0/0.5 | −2.4/−3.4/4.3/0.4/ 1.8/−0.4 | 88.87*** | CC = 0.2 |
| Own children (yes/no) | −5.2/4.9 | 4.7/−4.4 | 92.74*** | 2.36 (1.98, 2.82) |
| Smoker (yes/no) | −0.7/0.4 | 0.7/−0.4 | 1.35 | 1.12 (0.92, 1.36) |

**Notes:**

[1] Coding of relationship status: single, in a relationship, married, divorced, widowed, other. CC, Contingency Coefficient; CRT, Cognitive Reflection Test.

\* $p < 0.05$.

\*\* $p < 0.01$.

\*\*\* $p < 0.001$ (two-tailed).

**Table 3 Differences in CRT scores between experienced and naïve participants separated by highest educational qualification.**

|  |  | CRT score (%) | | | | |
|---|---|---|---|---|---|---|
|  | Mean CRT score (SD) | 0 | 1 | 2 | 3 | N |
| Not completed | 1.16 (1.08) | 36 | 25 | 25 | 14 | 44 |
|  | 1.22 (1.03) | 30 | 32 | 24 | 14 | 37 |
| Primary education | 0.93 (0.97) | 43 | 29 | 20 | 8 | 191 |
|  | 1.16 (1.17) | 43 | 16 | 23 | 18 | 122 |
| Apprenticeship diploma | 1.06 (1.03) | 39 | 27 | 23 | 11 | 359 |
|  | 1.50 (1.12) | 26 | 21 | 30 | 23 | 121 |
| Secondary education | 1.31 (1.05) | 29 | 27 | 28 | 16 | 344 |
|  | 1.65 (1.11) | 22 | 20 | 29 | 29 | 390 |
| University degree | 1.55 (1.07) | 23 | 23 | 32 | 22 | 236 |
|  | 1.99 (0.64) | 10 | 17 | 37 | 36 | 285 |

**Note:**

First-line entry = naïve participants, second-line entry = experienced participants.

correlation was below one ($r_{sp} = 0.76$, $p < 0.01$). Therefore, the dominance weights should be given preference over beta weights.

As can be seen in Table 4, the best predictors of the CRT score were the highest educational qualification, followed by CRT experience, participant's sex (male higher scores), and being a smoker (smokers had lower CRT scores, i.e. scored rather intuitive

**Table 4 Results of the linear regression and dominance analysis with the CRT score as the dependent measure.**

|  | $\beta$ | Zero-order correlation $r$ | Dominance $R^2$ (%) |
|---|---|---|---|
| Age | −0.018 | −0.028 | 0.05 |
| Education | 0.203*** | 0.239*** | 4.62 |
| Number of siblings | −0.001 | −0.048* | 0.08 |
| Sex | −0.138*** | −0.141*** | 1.88 |
| Own children | 0.011 | −0.026 | 0.03 |
| Smoker | −0.091*** | −0.088*** | 0.80 |
| CRT experience | 0.158*** | 0.198*** | 3.06 |
| Relationship status |  |  |  |
|    Single | −0.016 | −0.055** | 0.19 |
|    In a relationship | 0.077 | 0.084*** | 0.32 |
|    Married | 0.062 | 0.005 | 0.07 |
|    Divorced | 0.015 | −0.016 | 0.02 |
|    Widowed | −0.003 | −0.069** | 0.20 |

Notes:
$F(12,2070) = 22.03$, $p < 0.001$; adj. $R^2 = 10.8\%$.
Coding of Sex: 1..male, 2..female; Coding of 'own children' and smoker: 0..no, 1..yes.
* $p < 0.05$.
** $p < 0.01$.
*** $p < 0.001$ (two-tailed).

than reflective). Most of the overall explained variance could be attributed to the first two predictors (highest education qualification, CRT experience: 7.68% in total). We repeated the analyses for each CRT item, but the pattern of results remained constant. The only exception was that for the CRT Item 1, age was also significant (the lower the age, the higher the score; $\beta = −0.095$; detailed results omitted for brevity).

## DISCUSSION

In the present study we could clearly show that having prior experience with the CRT or any similar task has a substantial influence on the CRT score ($d = 0.41$). CRT experience was one of the best predictors of the CRT score (3.06% explained variance), along with the highest educational qualification (4.62%).

Experienced participants not only had a higher CRT score (~half a point on average), they were also different from naïve participants regarding their socio-demographics (e.g., young, lower educated, being rather single or in a relationship than married, and having no children). Furthermore, floor and ceiling effects were prevalent. Almost one third of higher educated (secondary education, university) experienced participants reached the highest possible score, whereas one third of the lower educated participants did not solve any CRT item at all.

This suggests that the CRT in its original three-item form (*Frederick, 2005*) is not only limited by familiarity, it is also limited by range restrictions. Although the items seem to be of medium difficulty, the classical CRT is not suitable for the highly educated (because they solve all items) as well as the lowly educated (because they solve none of them).

Therefore, we strongly recommend using recent multi-item CRTs (including new intuitive vs. reflection items; *Primi et al., 2015*; *Toplak, West & Stanovich, 2014*). This recommendation has several implications. First, following our recommendation should lower the probability that participants have experience with the CRT's items, at least for some time. When contrasting results based on old and new items, one can evaluate a possible influence from CRT experience (at least with the original CRT). If one wants to avoid the original CRT items entirely, *Toplak, West & Stanovich (2014)* also introduced a four-item version of the CRT including only new items. Second, multi-item CRTs have higher measurement reliability. Currently the CRT's reliability is rather low–in the present sample, naïve participants produced a rather low reliability of Cronbach $\alpha = 0.595$. Also, it could be that reliability estimates from past studies are impaired by CRT experience as well, i.e., the real reliability estimate is actually even lower. Using reliable multi-item CRTs should also raise the possibility to find substantial and meaningful effects in future research projects because increased measurement reliability leads to larger effect sizes (for a discussion, see *LeBel & Paunonen, 2011*) as well as less range restrictions.

The present results are also of interest regarding another effect–the Flynn effect (i.e., secular IQ gains; for an overview see *Williams, 2013*). The Flynn effect is the substantial increase in intelligence (fluid and crystallized) test scores over time (from ~1930 until now). The reasons for this effect are still under debate, but it seems to be a mixture of several influences such as real IQ gains through education or increases in test-specific skills (e.g., *Jensen, 1998*). For the CRT, the same effect of increasing test scores has been found by *Brañas-Garza, Kujal & Lenkei (2015)* in their cross-temporal meta-analysis. Initially, this could mean that people indeed are becoming more rational and reflective (compared to intuitive) over time. But the authors also found that this effect was driven by online samples, i.e., the effect might be driven by participants looking up the correct answers on the Internet. The present study extends and clarifies this supposition, that indeed the prevalence rate of knowing the CRT (and similar tasks) is high and indeed prior experience with the task substantially raises the CRT score. Hence, the gain in CRT scores is due to gains in test-specific skills rather than a gain in rational thinking.

## Limitations

In the present study only German-speaking volunteers were recruited, therefore it is unclear how the found results apply to other communities. We believe that for English-speaking countries the CRT experience effect might even be more prevalent. First of all, the CRT was originally published in English, so there was more time for items to become well-known in English. Second, as far as we know, the CRT items (especially the first bat-and-ball item) are frequently used in introductory courses and classroom presentations in the US and UK. In the future, it would be interesting to analyze potential experience effects with the CRT items in other languages (e.g., Spanish; *Gòmez-Chacòn et al., 2012*).

Furthermore, in the present study we only asked about whether or not participants have any experience with the CRT items or similar tasks, but we did not ask how much

experience they have and when they made this experience (long time ago vs. very recently). Assessing these two additional variables would make it possible do draw a more nuanced picture about the influence of CRT experience onto CRT scores.

## CONCLUSION

To sum up, we think that a methodologically well-developed CRT is vital in order to settle debates about the CRT (e.g., its dependency on numerical ability and correlation with intelligence). Currently the classical CRT (*Frederick, 2005*) is limited by familiarity because it is frequently found on the Internet and is frequently used in introductory courses and classroom presentations. Furthermore, it is limited by range restrictions due to its three-item form. With the present study we could show that experience strongly affects the CRT and a revalidation of the task is therefore indicated. Fortunately, there are already encouraging new developments regarding the CRT (e.g., *Primi et al., 2015*; *Toplak, West & Stanovich, 2014*) which should be given preference over the classical CRT.

### Funding
The authors received no funding for this work.

### Competing Interests
The authors declare that they have no competing interests.

### Author Contributions
- Stefan Stieger conceived and designed the experiments, performed the experiments, analyzed the data, contributed reagents/materials/analysis tools, wrote the paper, prepared figures and/or tables, reviewed drafts of the paper.
- Ulf-Dietrich Reips analyzed the data, reviewed drafts of the paper.

### Human Ethics
The following information was supplied relating to ethical approvals (i.e., approving body and any reference numbers):

The present study was conducted in accordance with the principles of the Declaration of Helsinki and with institutional guidelines of the School of Psychology, University of Vienna. Furthermore, the present study followed the Guidelines for ethical conduct of behavioral projects involving human participants proposed by the American Psychological Association. According to the institutional guidelines of the University of Vienna, Austria (http://satzung.univie.ac.at/ethikkommission-der-universitaet-wien/), approval by an ethics committee was not necessary because the study did not affect the physical or psychological integrity, the right for privacy, or other personal rights or interests (see §2(1)). All participants gave verbal informed consent after having received a written description of the study and could withdraw participation at any point. Data collection was anonymous and no harmful procedures were used.

## Data Deposition

Open Science Framework: https://osf.io/3mkhj/.

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
