# Peer review of "A limitation of the Cognitive Reflection Test: familiarity"

_PeerJ, doi:10.7717/peerj.2395_

## Round 0.1 · original submission · Minor Revisions

Dear Authors,

The first two peer reviewers have given important comments for your team to proceed to improve your manuscript so that it can be resubmitted for re review.

Reviewer 1 ·

Basic reporting

This paper will be an important addition to the existing literature on the CRT and would seem of considerable interest to those conducting research in the area of cognitive functioning. In addition, it is likely that the results will stimulate new thinking in the field.

The authors clearly define a research objectives and give a throughout introduction to the literature on the CRT. The research methodology is described in sufficient details and the statistical analysis is performed to a high technical standard. The paper has a good sample and has tested various regression approaches.

In what follows, I present some questions and issues that should be addressed in a revision. I believe that addressing them will make the paper more effective and highly cited.

Experimental design

1. Could you please clarify the sample collection procedure? I understand you recruited friends and relatives of research assistants (Line 113) but it it’s not clear how was the study conducted.

Did you invite the participants to the lab to run the study there or you approached them in their homes?

Also, is the sample nationally representative or participants are mainly from the town of the authors?

2. Since you are directly targeting friends and family this is not exactly a random sample. Is there a self-selection bias in your sample? Please, could you provide a discussion on whether these are valid concerns (and what can you do about them)?

3. As shown by the results experience with the test is an important determinant of the of CRT scores. However, it would be also important to see how many times they sat the test and when was the last time they’ve seen the CRT items. There might be non-students who have only seen it few years ago and don’t really remember the answers, but also university students who regularly participate in experiments in labs where researchers use the CRT.

What was the distribution in this current study of students/non-students among experienced subjects? Is it perhaps true that your results on experience were driven by university students participating in lab studies?

Validity of the findings

1. In Line 290 you mention that the reliability of the original CRT compared to the multi-item CRT is low. What do you mean by reliability in this context and how do you measure it?

2. Some researchers consider the CRT as a poor measure of pure cognition. Are there any alternative measures of cognitive abilities that you propose to be good substitutes of the CRT? You mention some of the extended versions of the CRT e.g. CRT7 by Toplak et. al (2014), but these all involve the original CRT items.

3. Are there any significant differences in terms of socio-demographic status and experience between those who gave an incorrect and intuitive answer and those who gave an incorrect but not intuitive answer?

4. Do your results hold if you run a regression on the groups with highest (CRT=3) or the lowest cognitive abilities (CRT=0) as dependent variables?

Additional comments

Overall, I think these findings are of great interest for economists and psychologists, among others. It will be an important addition to the existing literature on the CRT. Therefore I feel that this paper is an excellent fit for the general audience of PeerJ.

·

Basic reporting

In the introduction, it is necessary to substantiate better the theoretical motivation of the work, as well as the contribution of this study to the existing literature. Additionally I suggest to justify the measure of demographic variables and to describe the expected results (e.g smoking behavior).
Please note that the Primi et al. CRT final version is composed of 6 items. Add also the new reference (Primi, C., Morsanyi, K., Chiesi, F., Donati, M. & Hamilton, J. (2015). The Development and Testing of a New Version of the Cognitive Reflection Test Applying Item Response Theory (IRT) Journal of Behavioral Decision Making, doi: 10.1002/bdm.1883.)

Experimental design

I suggest to add in the table the percentage of missing answers for each item in both groups. and add a comment.

Validity of the findings

To better understand the differences between naïve participants and who have prior experience with the CRT, it’s necessary to add a sample matched for the others variables

Additional comments

The Author should comment more deeply on the pluses and limitations of the research. In particular, adding which is the contribution of this study to the existing literature on CRT.

Reviewer 3 ·

Basic reporting

I think it is a publishable work. No Comments!

Experimental design

The authors may consider reporting reliability test in this section.

Validity of the findings

No Comments

---

## Round 0.2 · accepted · Accept

Dear Authors,

Thank you for your hard work in re-submitting a revised manuscript which has been accepted.

Congratulations.

·

Basic reporting

No comments

Experimental design

No comments

Validity of the findings

No comments

Additional comments

I’m happy with the revised manuscript. I think that the paper in the revised version is a nice contribution in this research area.